# Multidimensional encoding of restricted and anisotropic diffusion by double rotation of the *q* vector

**Hong Jiang, Leo Svenningsson, and Daniel Topgaard**

Physical Chemistry, Lund University, P.O. Box 124, 22100 Lund, Sweden

**Correspondence:** Daniel Topgaard (daniel.topgaard@fkem1.lu.se)

**Abstract.** `CE1` Diffusion NMR and MRI methods building on the classic pulsed gradient spin-echo sequence are sensitive to many aspects of translational motion, including time and frequency dependence ("restriction"), anisotropy, and flow, leading to ambiguities when interpreting experimental data from complex heterogeneous materials such as living biological tissues. While the oscillating gradient technique specifically targets frequency dependence and permits control of the sensitivity to flow, tensor-valued encoding enables investigations of anisotropy in orientationally disordered materials. Here, we propose a simple scheme derived from the "double-rotation" technique in solid-state NMR to generate a family of modulated gradient waveforms allowing for comprehensive exploration of the 2D frequency–anisotropy space and convenient investigation of both restricted and anisotropic diffusion with a single multidimensional acquisition protocol, thereby combining the desirable characteristics of the oscillating gradient and tensor-valued encoding techniques. The method is demonstrated by measuring multicomponent isotropic Gaussian diffusion in simple liquids, anisotropic Gaussian diffusion in a polydomain lyotropic liquid crystal, and restricted diffusion in a yeast cell sediment.

## 1 Introduction

Magnetic field gradients applied during the dephasing and rephasing periods of a spin-echo sequence (Hahn, 1950) render the NMR signal sensitive to various aspects of translational motion including bulk diffusivity (Douglass and McCall, 1958), flow (Carr and Purcell, 1954), time and frequency dependence ("restriction") (Woessner, 1963), anisotropy (Boss and Stejskal, 1965), and exchange (Kärger, 1969). Although the conventional and ubiquitous pulsed gradient spin-echo sequence by Stejskal and Tanner (1965) may give information about all of these aspects, more elaborate gradient modulations (Tanner, 1979; Cory et al., 1990; Callaghan and Manz, 1994; Mori and van Zijl, 1995) are required to unambiguously assign a certain mechanism to the experimental observations (Topgaard, 2017; Lundell and Lasič, 2020). Diffusion MRI methods incorporating such advanced diffusion encoding schemes have recently been shown to have potential for clinical research applications (Reymbaut et al., 2020); some notable examples are oscillating gradients to estimate cell sizes (Xu et al., 2021) and tensor-valued encoding to characterize cell shapes (Daimiel Naranjo et al., 2021) in breast tumors.

The sensitivity of the MRI signal to the various types of motion can be quantified with the tensor-valued encoding spectrum $\mathbf{b}(\omega)$ (Topgaard, 2019a; Lundell and Lasič, 2020), the trace of which equals the dephasing power spectrum (Stepišnik, 1981) – relevant for isotropic restricted diffusion – and whose integral over $\omega$ equals the conventional $b$ matrix (Basser et al., 1994) giving information about diffusion anisotropy. While most studies focus on either the frequency-dependent (Aggarwal, 2020) or tensorial (Reymbaut, 2020) aspects of the encoding, Lundell et al. (2019) suggested joining them into a common multidimensional framework. The approach was demonstrated with gradient waveforms deriving from the magic-angle spinning (MAS) technique in solid-state NMR spectroscopy (Andrew et al., 1959; Eriksson et al., 2013; Topgaard, 2013); however, these methods offer only limited access to the frequency and anisotropy dimensions.

Expanding on the results of Lundell et al. (2019), we take inspiration from the "double-rotation" (DOR) technique in solid-state NMR (Samoson et al., 1998) and derive a family of gradient waveforms for comprehensive exploration of, in particular, the frequency–anisotropy dimensions of $\mathbf{b}(\omega)$, as quantified by the centroid frequency $\omega_{\text{cent}}$ (Arbabi et al., 2020) and encoding anisotropy $b_\Delta$ (Eriksson et al., 2015), in addition to the $b$ value and $b$ vector $(\Theta, \Phi)$ of conventional diffusion tensor imaging (Kingsley, 2006). While $\omega_{\text{cent}}$ is key for characterizing restricted diffusion (Stepišnik and Callaghan, 2000), the variable $b_\Delta$ enables quantification of anisotropy in orientationally disordered materials (Eriksson et al., 2015) and estimation of nonparametric diffusion tensor distributions (de Almeida Martins and Topgaard, 2016; Topgaard, 2019b). The ability of the new gradient waveforms to give access to the complete 2D $\omega_{\text{cent}}$–$b_\Delta$ plane is demonstrated by microimaging measurements on previously studied phantoms with well-defined restriction and anisotropy properties, namely water (Mills, 1973) and concentrated salt solution (Wadsö et al., 2009) with isotropic Gaussian diffusion, a lamellar liquid crystal giving anisotropic Gaussian diffusion (Topgaard, 2016), and a yeast cell sediment exhibiting isotropic restricted diffusion (Malmborg et al., 2006).

## 2   Theoretical background

To set the stage for later sections dealing with the proposed gradient waveforms to investigate both frequency-dependent and tensorial aspects of translational motion, we include a brief summary of the relevant theory here; greater detail can be found in the textbooks by Price (2009) and Callaghan (2011) as well as in the comprehensive review by Lundell and Lasič (2020). Readers already familiar with the background material may proceed directly to the design of gradient waveforms in Sect. 3 after noting Eqs. (32) and (34) with the definitions of the main variables $\omega_{\text{cent}}$ and $b_\Delta$ reporting on the sensitivity to restriction and anisotropy.

### 2.1   Encoding of translational motion by magnetic field gradients

Figure 1 illustrates the effects of a general gradient waveform $\mathbf{g}(t)$ on the NMR signal from an ensemble of spins undergoing restricted diffusion and flow within an infinite cylinder. As shown using the 2D and 3D plots of $\mathbf{g}(t)$ in Fig. 1a, both the magnitude and direction of the gradient vector are changing smoothly with time. Simultaneously, the spins spread out and gradually drift from their initial positions (Fig. 1b). The time-dependent normalized signal $E(t)$ is given by

$$E(t) = \langle \exp(i\phi(t)) \rangle, \tag{1}$$

where $\langle \ldots \rangle$ denotes an ensemble mean and the time-dependent phase $\phi(t)$ of a single spin with gyromagnetic ratio $\gamma$ is determined by the time integral of the scalar product between $\mathbf{g}(t)$ the time-dependent position $\mathbf{r}(t)$ according to

$$\phi(t) = -\gamma \int_0^t \mathbf{g}(t') \cdot \mathbf{r}(t') \, \mathrm{d}t'. \tag{2}$$

The interplay between $\mathbf{g}(t)$ and $\mathbf{r}(t)$ results in $\phi(t)$ evolving from zero for all spins at $t = 0$ to periodic patterns with varying directions and spatial wavelengths at intermediate times and, finally, an overall phase shift superposed on partially randomized values. From Eq. (1), it follows that the latter phase dispersion leads to a decrease in the magnitude of the signal.

The evolution of $\phi(t)$ may be rationalized by partial integration of Eq. (2) into

$$\phi(t) = -\mathbf{q}(t) \cdot \mathbf{r}(t) + \int_0^t \mathbf{q}(t') \cdot \mathbf{v}(t') \, \mathrm{d}t', \tag{3}$$

where $\mathbf{q}(t)$ is the dephasing vector, defined as

$$\mathbf{q}(t) = \gamma \int_0^t \mathbf{g}(t') \, \mathrm{d}t', \tag{4}$$

and $\mathbf{v}(t) = \mathrm{d}\mathbf{r}(t) / \mathrm{d}t$ is the time-dependent velocity. The spatial periodicity in $\phi(t)$ at intermediate times is, according to the first term in Eq. (3), given by the scalar product between $\mathbf{q}(t)$ and $\mathbf{r}(t)$, which is utilized to obtain the spatial resolution in MRI where the dephasing vector is usually denoted $\mathbf{k}(t)$. The plots of $\mathbf{q}(t)$ in Fig. 1a show smooth changes in both the magnitude and direction of the vector with time.

Focusing on translational displacements rather than absolute positions, we select a time $\tau$ where

$$\mathbf{q}(\tau) = 0 \tag{5}$$

and the first term in Eq. (3) vanishes while the second one remains:

$$\phi(\tau) = \int_0^\tau \mathbf{q}(t) \cdot \mathbf{v}(t) \, \mathrm{d}t. \tag{6}$$

The value of $\phi(\tau)$ in Eq. (6) is insensitive to $\mathbf{r}(\tau)$ but depends on the history of $\mathbf{q}(t)$ and $\mathbf{v}(t)$ in the interval from $t = 0$ to $\tau$.

### 2.2   Gaussian phase distribution approximation

As shown in Figure 1c, the selected gradient waveform and random walk simulation parameters yield a phase distribution $P(\phi(t))$ that is well approximated at $t = \tau$ as a Gaussian function with mean $\langle \phi(\tau) \rangle$ and standard deviation $\text{std}[\phi(\tau)] = (\langle \phi(\tau)^2 \rangle - \langle \phi(\tau) \rangle^2)^{1/2}$. The Gaussian function can be expressed as

$$P(\phi(\tau)) = \frac{1}{2\sqrt{\pi\beta}} \exp\left(-\frac{(\phi(\tau) - \alpha)^2}{4\beta}\right), \tag{7}$$

where

$$\alpha = \langle \phi(\tau) \rangle \tag{8}$$

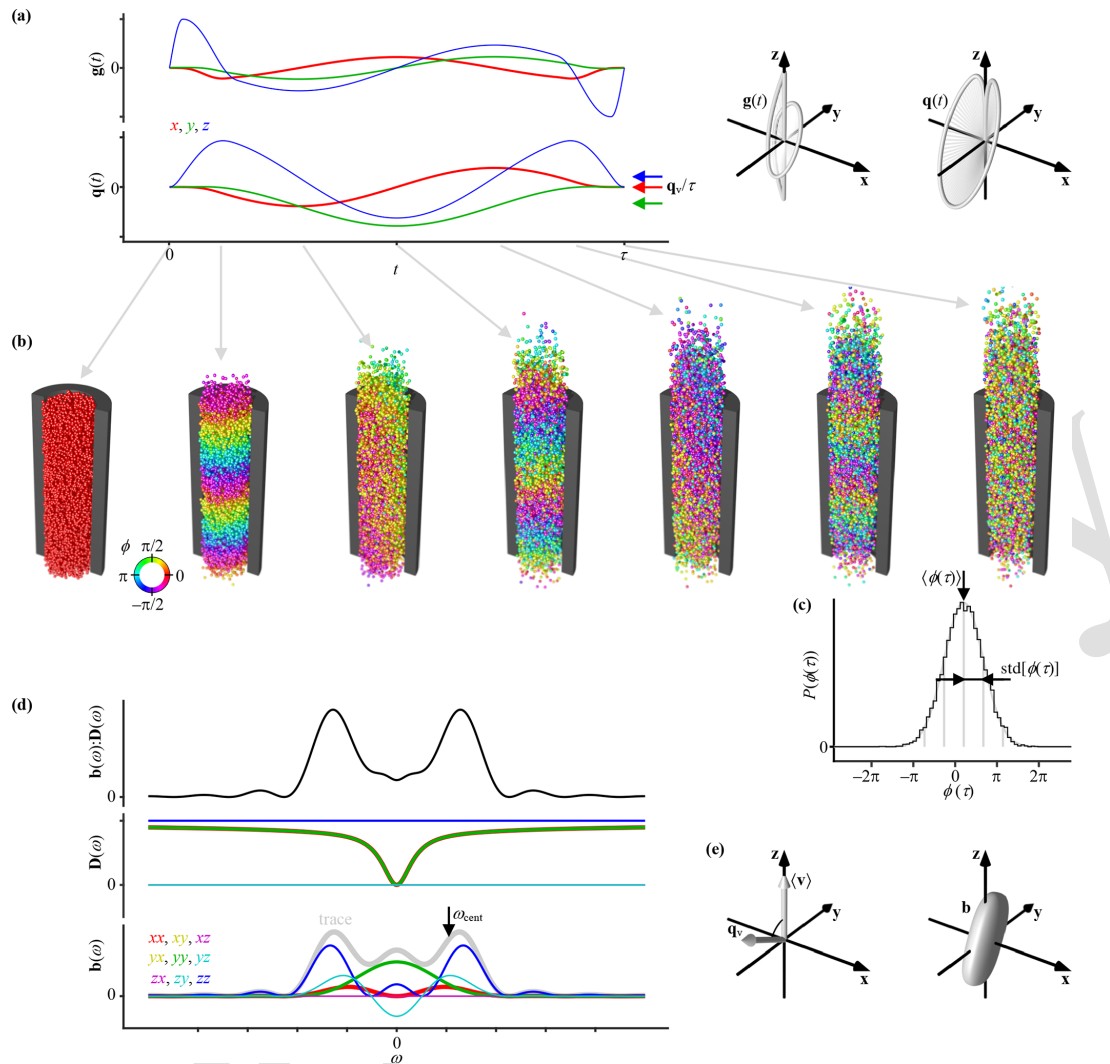

**Figure 1.** Principles of motion encoding by general gradient waveforms. **(a)** Time-dependent gradient $\mathbf{g}(t)$ and dephasing vector $\mathbf{q}(t)$ illustrated as 2D graphs of Cartesian components vs. $t$ (left) and 3D plots of trajectories through space (right). The average values of the $\mathbf{q}(t)$ components within the time interval from $t = 0$ to $\tau$ are indicated with arrows labeled $\mathbf{q}_V/\tau$, where the velocity-encoding vector $\mathbf{q}_V$ is defined in Eq. (15). **(b)** Sequence of snapshots from a random walk simulation of an ensemble of spins (spheres) undergoing restricted diffusion and flow within an infinite cylinder (black section) during application of the waveforms in panel (a). Each spin is color-coded by its time-dependent phase $\phi(t)$ given by the interplay between $\mathbf{g}(t)$ and the spin positions $\mathbf{r}(t)$ according to Eq. (2). **(c)** Phase distribution $P(\phi(\tau))$ for the ensemble of spins at $t = \tau$ (black histogram) and a Gaussian (smooth gray line) with mean $\langle\phi(\tau)\rangle$ and standard deviation $\mathrm{std}[\phi(\tau)]$ (spacing between vertical gray lines) which give the phase shift $\alpha$ and attenuation factor $\beta$ of the signal $E(\tau)$ via Eqs. (8), (9), and (11). **(d)** Frequency-dependent elements (color-coded) of the tensor-valued encoding spectrum $\mathbf{b}(\omega)$ and diffusion spectrum $\mathbf{D}(\omega)$ as well as their tensor dot product $\mathbf{b}(\omega) : \mathbf{D}(\omega)$ which gives $\beta$ via Eq. (21). The centroid frequency $\omega_{\mathrm{cent}}$ (arrow) is obtained from the trace of $\mathbf{b}(\omega)$ (gray line) via Eq. (32). **(e)** The 3D plots of the ensemble mean velocity $\langle\mathbf{v}\rangle$, velocity-encoding vector $\mathbf{q}_V$, and encoding tensor $\mathbf{b}$, with the latter being defined in Eq. (33). The scalar product of $\langle\mathbf{v}\rangle$ and $\mathbf{q}_V$ gives $\alpha$ via Eq. (14).

and

$$\beta = \frac{1}{2}\left(\left\langle\phi(\tau)^2\right\rangle - \langle\phi(\tau)\rangle^2\right). \tag{9}$$

After rewriting Eq. (1) as an integral,

$$E(\tau) = \int_{-\infty}^{\infty} P(\phi(\tau)) \exp(\mathrm{i}\phi(\tau)) \,\mathrm{d}\phi(\tau), \tag{10}$$

the insertion of Eq. (7) into Eq. (10) may be evaluated to

$$E(\tau) = \exp(\mathrm{i}\alpha - \beta), \tag{11}$$

where $\alpha$ and $\beta$ can be identified as quantitative measures of the overall phase shift and attenuation of the signal, respectively, as previously deduced from visual inspection of the phases of the spin ensemble in Fig. 1b. The Gaussian phase distribution approximation has been applied for the cases of

free (Carr and Purcell, 1954; Douglass and McCall, 1958) and restricted (Neuman, 1974) diffusion, and investigations of its ranges of validity can be found in the literature (Balinov et al., 1993; Stepišnik, 1999).

## 2.3 Mean velocity and velocity correlation function

Insertion of Eq. (6) into Eq. (8) yields

$$\alpha = \left\langle \int_0^\tau \mathbf{q}(t) \cdot \mathbf{v}(t) \, \mathrm{d}t \right\rangle, \tag{12}$$

which by separating $\mathbf{v}(t)$ into the ensemble mean $\langle \mathbf{v}(t) \rangle = \langle \mathbf{v} \rangle$ and fluctuating part $\mathbf{u}(t)$, defined by

$$\mathbf{u}(t) = \mathbf{v}(t) - \langle \mathbf{v} \rangle, \tag{13}$$

can be evaluated to

$$\alpha = \mathbf{q}_{\mathrm{v}} \cdot \langle \mathbf{v} \rangle, \tag{14}$$

where the flow encoding vector $\mathbf{q}_{\mathrm{v}}$ is defined as

$$\mathbf{q}_{\mathrm{v}} = \int_0^\tau \mathbf{q}(t) \, \mathrm{d}t. \tag{15}$$

Correspondingly, the insertion of Eq. (6) into Eq. (9) gives

$$\beta = \frac{1}{2} \left( \left\langle \left[ \int_0^\tau \mathbf{q}(t) \cdot \mathbf{v}(t) \, \mathrm{d}t \right]^2 \right\rangle - \left\langle \int_0^\tau \mathbf{q}(t) \cdot \mathbf{v}(t) \, \mathrm{d}t \right\rangle^2 \right), \tag{16}$$

which by reordering the time integrals and ensemble means as well as noting that $\langle \mathbf{u}(t) \cdot \langle \mathbf{v} \rangle \rangle = 0$, can be expressed as

$$\beta = \int_0^\tau \int_0^t \mathbf{q}(t)^{\mathrm{T}} \cdot \left\langle \mathbf{u}(t) \mathbf{u}(t')^{\mathrm{T}} \right\rangle \cdot \mathbf{q}(t') \, \mathrm{d}t \, \mathrm{d}t', \tag{17}$$

where $\langle \mathbf{u}(t) \mathbf{u}(t')^{\mathrm{T}} \rangle$ is the tensor-valued velocity correlation function.

## 2.4 Transformation to the frequency domain

After introducing the dephasing spectrum $q(\omega)$ and diffusion spectrum $\mathbf{D}(\omega)$ by Fourier transformations to the frequency ($\omega$) domain according to

$$\mathbf{q}(\omega) = \int_0^\tau \mathbf{q}(t) \exp(i\omega t) \, \mathrm{d}t \tag{18}$$

and

$$\mathbf{D}(\omega) = \frac{1}{2} \int_{-\infty}^\infty \left\langle \mathbf{u}(t) \mathbf{u}(t')^{\mathrm{T}} \right\rangle \exp\left( i\omega \left( t' - t \right) \right) \mathrm{d} \left( t' - t \right), \tag{19}$$

Eq. (17) can be recast into

$$\beta = \frac{1}{2\pi} \int_{-\infty}^\infty \mathbf{q}(\omega)^{\mathrm{T}} \cdot \mathbf{D}(\omega) \cdot \mathbf{q}(-\omega) \, \mathrm{d}\omega, \tag{20}$$

which can be expressed more compactly as

$$\beta = \int_{-\infty}^\infty \mathbf{b}(\omega) : \mathbf{D}(\omega) \, \mathrm{d}\omega, \tag{21}$$

where $\mathbf{b}(\omega)$ is the tensor-valued encoding spectrum defined as (Topgaard, 2019a; Lundell and Lasič, 2020)

$$\mathbf{b}(\omega) = \frac{1}{2\pi} \mathbf{q}(\omega) \mathbf{q}(-\omega)^{\mathrm{T}}, \tag{22}$$

and ":" denotes a tensor dot product (Basser et al., 1994):

$$\mathbf{b}(\omega) : \mathbf{D}(\omega) = \sum_i \sum_j b_{ij}(\omega) D_{ij}(\omega). \tag{23}$$

Combining Eqs. (11), (14), and (21) yields

$$E(\tau) = \exp\left( i\mathbf{q}_{\mathrm{v}} \cdot \langle \mathbf{v} \rangle - \int_{-\infty}^\infty \mathbf{b}(\omega) : \mathbf{D}(\omega) \, \mathrm{d}\omega \right), \tag{24}$$

where the motion-encoding properties of $\mathbf{g}(t)$ are summarized in $\mathbf{q}_{\mathrm{v}}$ and $\mathbf{b}(\omega)$, as illustrated in Fig. 1d and e. While $\mathbf{q}_{\mathrm{v}}$ and $\langle \mathbf{v} \rangle$ are ($\omega$-independent) vectors, both $\mathbf{b}(\omega)$ and $\mathbf{D}(\omega)$ are symmetric second-order tensors at each value of $\omega$.

## 2.5 Diffusion spectra for some simple cases

For a liquid with bulk diffusivity $D_0$ confined in $d$ dimensions in planar ($d = 1$), cylindrical ($d = 2$), or spherical ($d = 3$) compartments with radius $r$, the diffusion spectrum $D_{\mathrm{rest}}(\omega)$ in the restricted dimensions can be expressed as follows (Stepišnik, 1993):

$$D_{\mathrm{rest}}(\omega) = D_0 - \sum_k w_k \frac{D_0 - D_\infty}{1 + \omega^2 / \Gamma_k^2}, \tag{25}$$

where

$$\Gamma_k = \frac{\xi_k^2 D_0}{r^2} \tag{26}$$

and

$$w_k = \frac{2}{\xi_k^2 + 1 - d}. \tag{27}$$

Equation (25) includes the long-range diffusivity $D_\infty$, allowing for finite permeability of the compartment walls (Lasič et al., 2009), and can be recognized as a sum of Lorentzians with widths $\Gamma_k$ and weights $w_k$. In Eqs. (26) and (27), $\xi_k$ is the $k$th solution of

$$\xi J_{d/2-1}(\xi) - (d-1) J_{d/2}(\xi) = 0, \tag{28}$$

where $J_\nu$ is the $\nu$th-order Bessel function of the first kind. Using the cylindrical case in Fig. 1b as an example, the tensor-valued diffusion spectrum $\mathbf{D}(\omega)$ is given by

$$\mathbf{D}(\omega) = \mathbf{R}(\theta, \phi) \begin{pmatrix} D_{\mathrm{rest}}(\omega) & 0 & 0 \\ 0 & D_{\mathrm{rest}}(\omega) & 0 \\ 0 & 0 & D_0 \end{pmatrix} \mathbf{R}^{-1}(\theta, \phi), \tag{29}$$

where $\theta$ and $\phi$ are polar and azimuthal angles, giving the orientation of the cylinder in the lab frame, and $\mathbf{R}(\theta, \phi)$ is a rotation matrix. Figure 1d includes a plot of $\mathbf{D}(\omega)$ for the case $\theta = 0$ and $\phi = 0$ where all off-diagonal elements are zero. At high values of $\omega$, the diagonal elements converge towards $D_0$, corresponding to isotropic diffusion. Conversely, the effects of anisotropy reach a maximum in the low-$\omega$ limit where $D_{\text{rest}}(\omega)$ approaches $D_\infty$, which equals zero in the example in Fig. 1d. The planar version of Eq. (29) is obtained by exchanging $D_{\text{rest}}(\omega)$ and $D_0$. In the low-$\omega$ limit, the planar and cylindrical cases are often combined into a single expression:

$$\mathbf{D} = \mathbf{R}(\theta, \phi) \begin{bmatrix} D_\perp & 0 & 0 \\ 0 & D_\perp & 0 \\ 0 & 0 & D_\parallel \end{bmatrix} \mathbf{R}^{-1}(\theta, \phi), \tag{30}$$

where $D_\parallel$ and $D_\perp$ are the eigenvalues parallel and perpendicular to the main symmetry axis of the compartment, respectively. For completeness, we note that $D_{\text{rest}}(\omega)$ and $\mathbf{D}$ in Eqs. (25) and (30), respectively, reduce to an $\omega$-independent scalar diffusion coefficient $D$ for the special case of isotropic Gaussian diffusion where $D = D_0 = D_\infty = D_\perp = D_\parallel$.

## 2.6 Key properties of the tensor-valued diffusion encoding spectrum

While the signal expression in Eq. (24) takes the $\omega$ dependence and tensorial properties of both $\mathbf{b}(\omega)$ and $\mathbf{D}(\omega)$ into account and may be numerically evaluated as a single matrix multiplication after discretization in the $\omega$ dimension and appropriate reordering of the tensor elements, the common occurrence of systems exhibiting approximately Gaussian ($\omega$-independent) and/or isotropic diffusion has led to the introduction of simplified descriptions focusing on some specific aspects. In the absence of diffusion anisotropy – which is obviously not the case for the example in Fig. 1 – it is sufficient to use the dephasing power spectrum $b(\omega)$ (Stepišnik, 1981) obtained from $\mathbf{b}(\omega)$ by

$$b(\omega) = \text{trace}\{\mathbf{b}(\omega)\}. \tag{31}$$

The sensitivity to restriction can be summarized by the centroid frequency $\omega_{\text{cent}}$ (Arbabi et al., 2020), defined as

$$\omega_{\text{cent}} = \frac{1}{b} \int_{-\infty}^{\infty} |\omega| \, b(\omega) \, \mathrm{d}\omega. \tag{32}$$

In addition to all the tensor elements of $\mathbf{b}(\omega)$, Fig. 1d includes a plot of $b(\omega)$ with an arrow indicating $\omega_{\text{cent}}$. The example of $b(\omega)$ covers both low- and high-$\omega$ features of $\mathbf{D}(\omega)$ and is, thus, less well suited for exploring $\omega$-dependent diffusion processes than gradient modulation schemes comprising trains of rectangular pulses (Callaghan and Stepišnik, 1995), multiple smooth oscillations (Parsons et al., 2003), or a Carr–Purcell–Meiboom–Gill sequence in the presence of a constant gradient (Lasič et al., 2006), where the encoding

power is concentrated in a narrow frequency range and the single value $\omega_{\text{cent}}$ captures most of the relevant information about the spectral content. Even when the peaks in $b(\omega)$ are broader than the features in $\mathbf{D}(\omega)$, the $\omega_{\text{cent}}$ metric has some value as a bookkeeping tool but is less suitable for quantitative analysis.

For anisotropic systems with Gaussian diffusion, it is useful to introduce the $b$-matrix $\mathbf{b}$ (Basser et al., 1994) by

$$\mathbf{b} = \int_{-\infty}^{\infty} \mathbf{b}(\omega) \, \mathrm{d}\omega. \tag{33}$$

The sensitivity to anisotropy is given by the "shape" of the tensor which can be quantified with the encoding anisotropy $b_\Delta$ and asymmetry $b_\eta$ (Eriksson et al., 2015), defined as

$$b_\Delta = \frac{1}{b}\left(b_{ZZ} - \frac{b_{YY} + b_{XX}}{2}\right) \tag{34}$$

and

$$b_\eta = \frac{3}{2} \frac{b_{YY} - b_{XX}}{b \, b_\Delta}, \tag{35}$$

respectively. Here, $b_{XX}$, $b_{YY}$, and $b_{ZZ}$ are the eigenvalues of $\mathbf{b}$ ordered according to the Haeberlen convention $|b_{ZZ} - b/3| > |b_{XX} - b/3| > |b_{YY} - b/3|$ (Haeberlen, 1976) and

$$b = \text{trace}\{\mathbf{b}\} \tag{36}$$

is the conventional $b$ value (Le Bihan et al., 1986) that gives the overall magnitude of the diffusion encoding. Figure 1e shows a superquadric tensor glyph (Kindlmann, 2004) representation of $\mathbf{b}$ where all eigenvalues and both shape parameters $b_\Delta$ and $b_\eta$ are nonzero.

From the definitions of $\mathbf{q}_v$, $\mathbf{q}(\omega)$, and $\mathbf{b}(\omega)$ in Eqs. (15), (18), and (22), it follows that the $\omega = 0$ values of $\mathbf{b}(\omega)$ are proportional to $\mathbf{q}_v \mathbf{q}_v^{\mathrm{T}}$ and, thus, report on the sensitivity to flow, albeit with some ambiguity with respect to the directionality: the vectors $\mathbf{q}_v$ and $-\mathbf{q}_v$ give the same $\mathbf{b}(\omega = 0)$. It should be noted that $\mathbf{q}_v$ is not necessarily colinear with any of the eigenvectors of $\mathbf{b}$.

## 2.7 Special cases for data analysis

For data-fitting purposes, it is convenient to write the normalized signal $E$ as the ratio

$$E = S/S_0, \tag{37}$$

where $S$ is detected signal and $S_0$ is the signal obtained in a reference measurement with the amplitudes of the motion-encoding gradients set to zero. Equation (24) can then be expressed as

$$S = S_0 \exp\left(i\mathbf{q}_v \cdot \langle \mathbf{v} \rangle - \int_{-\infty}^{\infty} \mathbf{b}(\omega) : \mathbf{D}(\omega) \, \mathrm{d}\omega\right). \tag{38}$$

For the special cases of (i) isotropic restricted, (ii) anisotropic Gaussian, and (iii) isotropic Gaussian diffusion in the absence of net flow ($\langle \mathbf{v} \rangle = 0$), Eq. (38) is simplified to

$$S = S_0 \exp\left(-\int_{-\infty}^{\infty} b(\omega) \, D(\omega) \, \mathrm{d}\omega\right), \tag{39}$$

$$S = S_0 \exp(-\mathbf{b} : \mathbf{D}), \tag{40}$$

and

$$S = S_0 \exp(-bD), \tag{41}$$

respectively, where $D(\omega)$ is the (isotropic) diffusion spectrum, $\mathbf{D}$ is the ($\omega$-independent) diffusion tensor, and $D$ is the (isotropic and $\omega$-independent) diffusion coefficient introduced in Sect. 2.5 above.

For a heterogeneous system including multiple subensembles $i$ with individual signals $S_i$, each of which is given by one of the equations (Eqs. 38–41) above, the total signal $S$ is obtained by the sum

$$S = \sum_i S_i. \tag{42}$$

An important type of heterogeneity refers to the orientations of anisotropic objects with the extreme case of completely random orientations as in a "powder". In the special case of axial symmetry of both $\mathbf{b}$ and $\mathbf{D}$, powder averaging of Eq. (40) yields (Eriksson et al., 2015)

$$S = S_0 \exp(-bD_{\mathrm{iso}}) \frac{\sqrt{\pi}}{2} \frac{\exp(A/3)}{\sqrt{A}} \mathrm{erf}\left(\sqrt{A}\right), \tag{43}$$

where

$$A = 3bD_{\mathrm{iso}}b_\Delta D_\Delta. \tag{44}$$

In Eq. (44), $D_{\mathrm{iso}}$ is the isotropic diffusivity and $D_\Delta$ is the normalized diffusion anisotropy; these terms are defined as

$$D_{\mathrm{iso}} = \frac{1}{3}\mathrm{trace}\{\mathbf{D}\} = \frac{D_\parallel + 2D_\perp}{3} \tag{45}$$

and

$$D_\Delta = \frac{D_\parallel - D_\perp}{3D_{\mathrm{iso}}}, \tag{46}$$

respectively. Here, $D_\parallel$ and $D_\perp$ were introduced in Eq. (30). The definitions of $b_\Delta$ and $b$ can be found in Eqs. (34) and (36), respectively.

## 3 Design of gradient waveforms by double rotation of the *q* vector

Expanding on previous magic-angle spinning (Andrew et al., 1959; Eriksson et al., 2013; Topgaard, 2013) and variable-angle spinning (Frydman et al., 1992; Topgaard, 2016,

2017) approaches for generating motion-encoding gradient waveforms, we apply the double-rotation (DOR) technique (Samoson et al., 1998; Topgaard, 2019a) to probe the 2D acquisition space spanned by the variables $\omega_{\mathrm{cent}}$ and $b_\Delta$ defined in Eqs. (32) and (34), respectively. The $q$-vector trajectory $\mathbf{q}(t)$ is expressed in terms of its time-dependent magnitude $q(t)$ and unit vector $\mathbf{u}(t)$ as

$$\mathbf{q}(t) = q(t)\mathbf{u}(t). \tag{47}$$

For the special case of DOR, the unit vector is written as

$$\mathbf{u}(t) = \mathbf{R}_z(\psi_2(t))\mathbf{R}_y(\zeta_2)\mathbf{R}_z(\psi_1(t))\mathbf{R}_y(\zeta_1)\left[\begin{array}{ccc} 0 & 0 & 1 \end{array}\right]^{\mathrm{T}}, \tag{48}$$

where $\mathbf{R}_z$ and $\mathbf{R}_y$ are Euler rotation matrices, $\zeta_1$ and $\zeta_2$ are the inclinations of the two rotation axes, and $\psi_1(t)$ and $\psi_2(t)$ are the time-dependent angles of rotation. The rotations in Eq. (48) are applied from right to left and follow a $Z$–$Y$ active rotation matrix convention.

Starting from a conventional 1D gradient waveform $g_{1\mathrm{D}}(t)$ – for instance, a pair of rectangular or sine-bell pulses of opposite polarity – the time-dependent functions $q(t)$ and $\psi_2(t)$ are given by Topgaard (2016):

$$q(t) = \gamma \int_0^t g_{1\mathrm{D}}(t') \, \mathrm{d}t' \tag{49}$$

and

$$\psi_2(t) = \frac{\Delta\psi_2}{b} \int_0^t q^2(t') \, \mathrm{d}t', \tag{50}$$

where $\Delta\psi_2$ is the total angle of rotation during the encoding interval from time $t = 0$ to $\tau$ and

$$b = \int_0^\tau q^2(t) \, \mathrm{d}t \tag{51}$$

is the conventional $b$ value. After some exercises in trigonometry, combination of Eqs. (47)–(51) and the relation between $\mathbf{g}(t)$ and $\mathbf{q}(t)$ in Eq. (4) yields

$$\mathbf{g}_{\mathrm{DOR}}(t) = g_{1\mathrm{D}}(t)\begin{pmatrix} a_+ \cos\psi_+(t) + a_- \cos\psi_-(t) \\ +a_2 \cos\psi_2(t) \\ a_+ \sin\psi_+(t) - a_- \sin\psi_-(t) \\ +a_2 \sin\psi_2(t) \\ a_0 - a_1 \cos\psi_1(t) \end{pmatrix}$$
$$+ g_{\mathrm{rot}}(t)\begin{pmatrix} -(n+1)a_+ \sin\psi_+(t) \\ -(n-1)a_- \sin\psi_-(t) \\ -a_2 \sin\psi_2(t) \\ (n+1)a_+ \cos\psi_+(t) \\ -(n-1)a_- \cos\psi_-(t) \\ +a_2 \cos\psi_2(t) \\ na_1 \sin\psi_1(t) \end{pmatrix}, \tag{52}$$

where

$$g_{\mathrm{rot}}(t) = \frac{\Delta\psi_2 q(t)^3}{\gamma b} \tag{53}$$

is the time-dependent magnitude of the rotating gradient vector;

$$\psi_1(t) = n\psi_2(t) \text{ and}$$
$$\psi_\pm(t) = (n \pm 1)\psi_2(t) \tag{54}$$

are time-dependent rotation angles; and

$$a_0 = \cos\zeta_1 \cos\zeta_2,$$
$$a_1 = \sin\zeta_1 \sin\zeta_2,$$
$$a_2 = \cos\zeta_1 \sin\zeta_2, \text{ and}$$
$$a_\pm = \sin\zeta_1 \frac{\cos\zeta_2 \pm 1}{2} \tag{55}$$

are amplitudes of the oscillating terms.

In the solid-state NMR field, DOR is applied with the inclinations $\zeta_1 = 54.7°$ and $\zeta_2 = 30.6°$, corresponding to zeros of the second and fourth Legendre polynomial, to eliminate both first- and second-order quadrupolar broadening of the NMR spectra of nuclei such as $^{23}$Na (Samoson et al., 1998). While the encoding anisotropy $b_\Delta$ in Eq. (34) is closely related to the second Legendre polynomial (Eriksson et al., 2015), we are not yet aware of any diffusion analog of the quadrupolar interactions involving the fourth Legendre polynomial. Instead, we found that DOR with the inclinations $\zeta_1 = 90°$ and $\zeta_2 = -54.7°$ yields desirable properties for diffusion encoding, namely spectral content concentrated to a narrow frequency window for all of the elements of $\mathbf{b}(\omega)$. For $n > 1$ and the special case of $g_{1D}(t) \propto [\delta(t) - \delta(t - \tau)]$, where $\delta(x)$ is the Dirac delta function, these angles yield an isotropic $b$ tensor, corresponding to $b_\Delta = 0$. Waveforms for any values of $b_\Delta$ and $b_\eta$ are then conveniently obtained by scaling the components of $\mathbf{g}_{DOR}(t)$ according to

$$\mathbf{g}(t) = \begin{bmatrix} g_X(t) \\ g_Y(t) \\ g_Z(t) \end{bmatrix} = \begin{bmatrix} g_{DOR,X}(t)\sqrt{1 - b_\Delta(1 + b_\eta)} \\ g_{DOR,Y}(t)\sqrt{1 - b_\Delta(1 - b_\eta)} \\ g_{DOR,Z}(t)\sqrt{1 + 2b_\Delta} \end{bmatrix}. \tag{56}$$

At the selected inclinations, the $a_0$ and $a_2$ terms in Eq. (52) equal zero, while the remaining amplitudes evaluate to $a_1 \approx -0.816$, $a_+ \approx 0.789$, and $a_- \approx -0.211$. For the special case $g_{1D}(t) \propto [\delta(t) - \delta(t - \tau)]$, the main frequency components of $\mathbf{b}(\omega)$ are thus given by

$$\omega_\pm = \frac{\psi_\pm(\tau)}{\tau} = (n \pm 1)\frac{\Delta\psi_2}{\tau} \text{ and}$$
$$\omega_1 = \frac{\psi_1(\tau)}{\tau} = n\frac{\Delta\psi_2}{\tau}, \tag{57}$$

where, according to Eq. (52), $\omega_\pm$ and $\omega_1$ are cleanly separated into the respective transverse ($X$, $Y$) and longitudinal ($Z$) directions. The mean frequency content, as quantified by the centroid frequency $\omega_{cent}$ defined in Eq. (32), can be estimated by weighting the contributions from the main frequency components by the corresponding amplitudes in

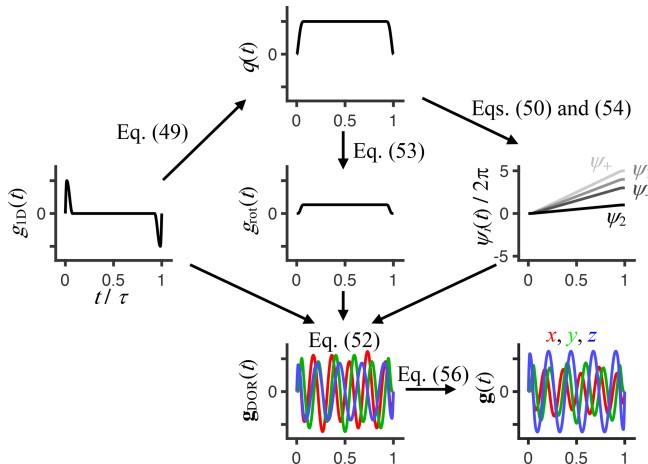

**Figure 2.** Flowchart for calculating a double-rotation gradient waveform $\mathbf{g}_{DOR}(t)$ given a 1D dephasing/rephasing waveform $g_{1D}(t)$, rotation axis inclinations $\zeta_1$ and $\zeta_2$, double-rotation ratio $n$, and total angle of rotation $\Delta\psi_2$ during the waveform duration $\tau$. The waveform $g_{1D}(t)$, containing dephasing and rephasing pulses with sinusoidal ramps of durations $\varepsilon_{up}$ and $\varepsilon_{down}$, gives the time-dependent magnitude of the dephasing vector $\mathbf{q}(t)$ via Eq. (49), which yields the time-dependent rotation angles $\psi_i(t)$ via Eqs. (50) and (54) as well as the time-dependent magnitude of the rotating and oscillating gradient $g_{rot}(t)$ via Eq. (53). Combining $g_{1D}(t)$, $g_{rot}(t)$, and $\psi_i(t)$ via Eq. (52) gives $\mathbf{g}_{DOR}(t)$, which, if $\zeta_1 = 90°$ and $\zeta_2 = -54.7°$, $n$ is an integer above 1, and $\Delta\psi_2$ is a multiple of $2\pi$ TS2, achieves isotropic encoding tensors $\mathbf{b}$ where the anisotropy $b_\Delta$ and asymmetry $b_\eta$ are both equal to zero. Finally, waveforms $\mathbf{g}(t)$ for any values of $b_\Delta$ and $b_\eta$ are obtained by scaling of the Cartesian components of $\mathbf{g}_{DOR}(t)$ according to Eq. (56). The shown example was generated with the MATLAB code provided in the Supplement using $\varepsilon_{up} = 0.015\tau$, $\varepsilon_{down} = 0.06\tau$, $\Delta\psi_2 = 2\pi$ TS3, $n = 4$, $b_\Delta = 0.5$, and $b_\eta = 0.25$.

Eq. (52) but is more accurately calculated by the numerical evaluation of Eq. (32), which also takes the finite durations of the sinusoidal oscillations into account. For rough prediction of $\omega_{cent}$, it is useful to note that $a_+^2 \gg a_-^2$, implying that the $\omega_+$ component will dominate the spectra in the $X$ and $Y$ directions. The scaling of the waveforms according to Eq. (56) preserves the frequency content in each of the eigendirections of the $b$ tensor but shifts the value of $\omega_{cent}$ between the approximate extremes $\omega_+$ and $\omega_1$ for $b_\Delta = -1/2$ and 1, respectively. The differences in $\omega_\pm$ and $\omega_1$ may give rise to a directional dependence of the sensitivity to restriction as investigated for the $b_\Delta = 0$ case by de Swiet and Mitra (1996).

Figure 2 illustrates the series of calculations required to convert a conventional 1D waveform $g_{1D}(t)$ and given values of $\zeta_1$, $\zeta_2$, $\Delta\psi_2$, $n$, $b_\Delta$, and $b_\eta$ to a 3D waveform $\mathbf{g}(t)$ by numerical evaluation of Eqs. (49)–(56). Following previous works to generate families of smooth gradient waveforms to explore the $b_\Delta$ and $b_\eta$ dimensions of diffusion encoding (Topgaard, 2016, 2017), we construct $g_{1D}(t)$ from a dephasing lobe with quarter-sine ramp-up of duration $\varepsilon_{up}$ and half-

cosine ramp-down of duration $\varepsilon_{\mathrm{down}}$ as well as a rephasing lobe obtained by inversion and time-reversal of the dephasing one. The corresponding MATLAB code is provided in the Supplement.

Figure 3 compiles waveforms and encoding spectra for an array of $n$ and $b_\Delta$ at constant $g_{\mathrm{1D}}(t)$, $\tau$, and $\Delta\psi_2$, yielding constant $b$. Increasing $n$ leads to larger rotation angles $\psi_1(t)$ and $\psi_\pm(t)$ and frequencies $\omega_1$ and $\omega_\pm$ according to Eqs. (54) and (57), respectively, at the expense of overall higher gradient amplitudes on account of the terms including $n$ in Eq. (52). For most waveforms, vanishing values of $\mathbf{b}(\omega)$ at $\omega = 0$ correspond to $\mathbf{q}_\mathrm{v} = 0$ and insensitivity to flow. Many of the examples in Fig. 3 are familiar from the literature, such as conventional Stejskal–Tanner encoding at ($n = 0$, $b_\Delta = 1$), basic flow-compensated encoding (Caprihan and Fukushima, 1990) at ($n = 1$, $b_\Delta = 1$), and magic-angle spinning of the $q$ vector (Eriksson et al., 2013) at ($n = 0$, $b_\Delta = 0$). The series of $b_\Delta = 1$ and $-1/2$ waveforms with varying $n$ resemble the cosine-modulated oscillating gradients of Parsons et al. (2003) and the circularly polarized version introduced by Lundell et al. (2015), respectively. Correspondingly, the series of waveforms with $n = 0$ and varying $b_\Delta$ has previously been introduced as a diffusion version of the variable-angle spinning technique to correlate isotropic and anisotropic chemical shifts in solid-state NMR (Topgaard, 2016, 2017). The approach for joint investigation of restricted and anisotropic diffusion proposed by Lundell et al. (2019), combining isotropic encoding with "tuned" and "detuned" directional encodings, can be recognized as measurements at the three discrete points ($n = 0$, $b_\Delta = 0$), ($n = 0$, $b_\Delta = 1$), and ($n = 1$, $b_\Delta = 1$) of the 2D plane in Fig. 3. For completeness, we note that the elliptically polarized oscillating gradients by Nielsen et al. (2018) can be reproduced with $b_\Delta = -1/2$ and $b_\eta$ in the range from 0 to 3 (not included in Fig. 3).

## 4   Proof-of-principle experiments

Magnesium nitrate hexahydrate, cobalt nitrate hexahydrate, and 1-decanol were purchased from Sigma-Aldrich Sweden AB and sodium octanoate was obtained from J&K Scientific via Th. Geyer in Sweden. Water was purified with a Milli-Q system. A sample with two-component isotropic diffusion was prepared by inserting a 4 mm NMR tube containing an aqueous solution saturated with magnesium nitrate (Wadsö et al., 2009) into a 10 mm NMR tube with water (Mills, 1973). The magnesium nitrate solution was spiked with a small amount of cobalt nitrate (0.27 wt % saturated solution) to reduce $T_1$ and $T_2$ to approx. 500 and 50 ms, respectively. An anisotropic sample was prepared by mixing 85.79 wt % Milli-Q, 9.17 wt % 1-decanol, and 5.04 wt% sodium octanoate giving a lamellar liquid crystal (Persson et al., 1975). Investigation of isotropic restricted diffusion was performed with a sediment of fresh baker's yeast (trade

name: Kronjäst; obtained from a local supermarket) prepared by dispersing yeast in tap water (1 : 1 volume ratio) in a glass vial, transferring it with a syringe to a 10 mm NMR tube to a sample height of 40 mm, and keeping the tube in an upright position overnight at 4 °C to allow the cells to settle under the force of gravity into a 20 mm high pellet (Malmborg et al., 2006).

MRI was performed on a Bruker AVANCE NEO 500 MHz spectrometer equipped with an 11.7 T vertical bore magnet and a MIC-5 microimaging probe fitted with a 10 mm radiofrequency insert for observation of ${}^1\mathrm{H}$. Images were acquired with a TopSpin 4.0 implementation of a spin-echo prepared single-shot RARE (Rapid Imaging with Refocused Echoes) sequence (available at https://github.com/daniel-topgaard/md-dmri, last access: 1 October 2022) using a $0.6 \times 0.6\,\mathrm{mm}^2$ resolution in a plane perpendicular to the tube axis, a 1 mm slice thickness, and a $16 \times 16 \times 1$ matrix size. Diffusion encoding employed pairs of identical gradient waveforms bracketing the 180° pulse in the preparation block (Lasič et al., 2014). Data were acquired for 8 $b$ values up to $6.44 \cdot 10^9\,\mathrm{s\,m^{-2}}$ and 15 orientations ($\Theta$, $\Phi$) for each of the 24 waveforms spanning the 2D $\omega_{\mathrm{cent}}$–$b_\Delta$ plane in Fig. 3 using a maximum gradient amplitude of $3\,\mathrm{T\,m^{-1}}$ and a waveform duration of $\tau = 25\,\mathrm{ms}$, giving values of $\omega_{\mathrm{cent}}$ in the range of 20–260 Hz. With a 5 s recycle delay, the total measurement time was approximately 4 h for each sample. The sample temperature was controlled with a Bruker VT unit: 278 K for the yeast and 291 K for the isotropic solutions and liquid crystal. For the yeast, the image slice was placed in the middle of the pellet and 10 mm below the bottom of the supernatant. Image reconstruction, definition of regions of interest, and curve fitting were performed in MATLAB using in-house code available at https://github.com/daniel-topgaard/md-dmri, last access: 1 October 2022 (Nilsson et al., 2018).

Figure 4 compiles experimental data and fits for all investigated samples. To facilitate visual inspection of the highly multidimensional data acquired as a function of ($b$, $\omega_{\mathrm{cent}}$, $b_\Delta$, $\Theta$, $\Phi$), the signal data were averaged over $b$-tensor orientations ($\Theta$, $\Phi$) and are displayed as conventional Stejskal–Tanner plots of $\log_{10}(S)$ vs. $b$ with the $\omega_{\mathrm{cent}}$ and $b_\Delta$ dimensions coded using different marker styles and a gray scale. For the isotropic Gaussian sample in Fig. 4a, all data points collapse onto a single master curve, thereby verifying that all 24 waveforms spanning the 2D $\omega_{\mathrm{cent}}$–$b_\Delta$ plane in Fig. 3 indeed give the same $b$ value. The pronounced nonlinearity of the $\log_{10}(S)$ vs. $b$ plot indicates the presence of multiple species with different diffusivities, and the bi-exponential fit yields diffusivities consistent with pure water (fast) and water in the saturated magnesium nitrate solution (slow). The anisotropic Gaussian phantom Fig. 4b yields data points stratified into one master curve for each of the four values of $b_\Delta$, verifying independence of $\omega_{\mathrm{cent}}$. The data are well fitted by the expression for randomly oriented axisymmetric diffusion tensors in Eq. (43), giving estimates of the diffusivities $D_\parallel$ and $D_\perp$ parallel and perpendicular to the cylindrical sym-

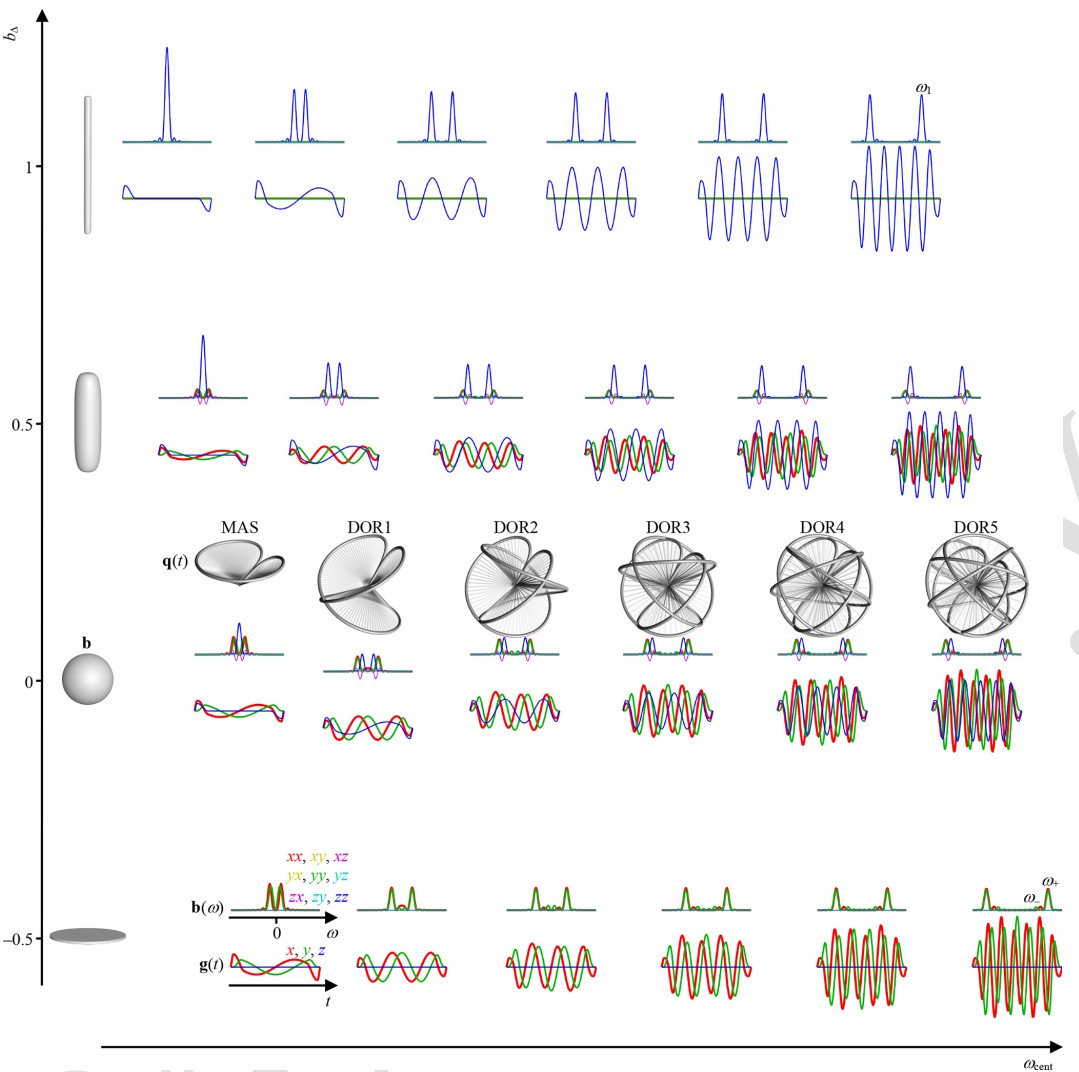

**Figure 3.** Gradient waveforms $g(t)$ for comprehensive exploration of the 2D space of centroid frequency $\omega_{cent}$ and anisotropy $b_\Delta$ of the tensor-valued encoding spectrum $\mathbf{b}(\omega)$. Magic-angle spinning (MAS) and double rotation (DOR$n$) with variable frequency ratio $n$ give $q$-vector trajectories $\mathbf{q}(t)$ shown as 3D plots for the $b_\Delta = 0$ cases. The waveforms are generated according to the scheme in Fig. 2 using $\varepsilon_{up} = 0.03\tau$, $\varepsilon_{down} = 0.12\tau$, $\Delta\psi_2 = 2\pi$ TS4, $b_\eta = 0$, and identical $b$ values for a 2D array of $n = 0, 1, \ldots, 5$ and $b_\Delta = -0.5, 0, 0.5$, and 1 with the angles $\zeta_1 = 0°$ and $\zeta_2 = 54.7°$ for $n = 0$ and $\zeta_1 = 90°$ and $\zeta_2 = -54.7°$ for $n>0$. Superquadric tensor glyphs (Kindlmann, 2004) along the vertical axis illustrate $\mathbf{b}$ for the chosen values of $b_\Delta$. The main maxima in $\mathbf{b}(\omega)$ are located at the frequencies $\omega_1$ and $\omega_+$ given in Eq. (57). Values of $\omega_{cent}$ and $b_\Delta$, including non-idealities originating from the finite durations of the dephasing and rephasing lobes of $g_{1D}(t)$, are obtained via Eqs. (32) and (34), respectively, using numerically evaluated $\mathbf{b}(\omega)$ according to Eqs. (4), (18), and (22).

metry axis of the crystallites. The observations $D_\parallel \ll D_\perp$ and $D_\parallel \approx 0$ are consistent with diffusion in a lamellar liquid crystal with planar surfactant bilayers being nearly impermeable to water (Callaghan and Söderman, 1983). For the isotropic restriction phantom in Fig. 4c, the signal depends strongly on $\omega_{cent}$. In this case, there is no clear stratification of data points into separate master curves on account of the interplay between $b_\Delta$ and $\omega_{cent}$, as reported in the legend in Fig. 4a and explained in detail below Eq. (57). The minor dependence of $\omega_{cent}$ on $b_\Delta$ is admittedly a drawback of our current approach for generating waveforms; however, we be-

lieve our approach is justified by the simplicity and transparency of the mathematical expressions in Eqs. (49)–(57). The sum of isotropic restricted and Gaussian components, given by Eqs. (41) and (39), yields an excellent fit to the acquired data, showing that the data feature no dependence on the value of $b_\Delta$. To account for the fact that $b(\omega)$ for (in particular) the $b_\Delta = 0$ waveforms cannot be well approximated with a delta function at a single value of $\omega$, the signal for the restricted component was obtained by numerical evaluation of the integral in Eq. (39) using the diffusion spectrum $D(\omega)$ for spherical compartments in Eq. (25). The obtained

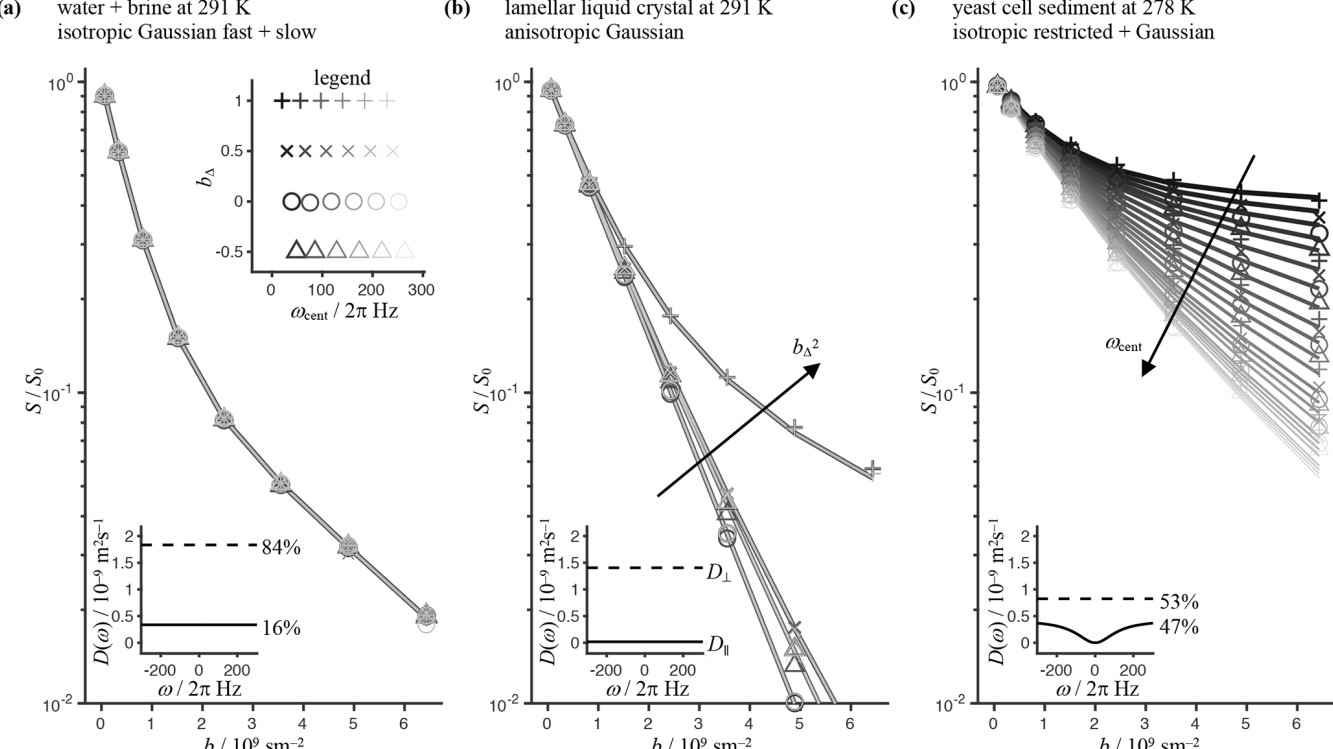

**Figure 4.** Experimental (markers) and fitted (lines) normalized powder-averaged signal $S/S_0$ vs. $b$ value for phantoms with well-defined diffusion properties. **(a)** Tube-in-tube assembly of pure water and a concentrated solution of magnesium nitrate in water ("brine") giving rise to two isotropic Gaussian ($\omega$-independent) components. A two-component fit based on Eq. (41) gives diffusion spectra $D(\omega)$, as shown in the inset, with percentages indicating the relative contributions. The legend shows the 24 investigated values in the 2D $\omega_{cent}$ (the gray scale) and $b_\Delta$ (marker style) space corresponding to the gradient waveforms in Fig. 3 with a duration of $\tau = 25$ ms, a maximum gradient strength of $3\,\mathrm{Tm^{-1}}$, and pairs of waveforms bracketing the 180° pulse in the spin-echo preparation. **(b)** Polydomain lamellar liquid crystal giving Gaussian parallel and perpendicular diffusivities, $D_\parallel$ and $D_\perp$, as estimated by a fit of Eq. (43). **(c)** Sediment of yeast cells with intra- and extracellular compartments, with the former exhibiting restricted ($\omega$-dependent) diffusion. The inset shows $D(\omega)$ resulting from a two-component fit with one spherically restricted (solid) and one Gaussian (dashed) component. For the former component, the signal was obtained by numerical integration of Eq. (39) with $D(\omega)$ given by Eq. (25) with $d = 3$ and $D_\infty$ constrained to zero.

diffusivities are consistent with previous results for intra- and extracellular water in yeast cell sediments (Åslund and Topgaard, 2009). Taken together, the data in Fig. 4 verify that the set of waveforms allows detailed exploration of the 2D $\omega_{cent}$–$b_\Delta$ plane of multidimensional diffusion encoding.

## 5   Conclusions and outlook

The proposed family of double-rotation gradient waveforms enables comprehensive sampling of both the frequency and "shape" dimensions of diffusion encoding, as required for detailed characterization of restriction and anisotropy in heterogeneous materials such as brain tissues. The present waveforms, deriving from simple geometrical considerations and generated by compact mathematical expressions, are suitable for preclinical investigations of tissue samples or small animals on high-gradient systems. By numerical optimizations to maximize the $b$ value for given gradient strength (Topgaard, 2013; Sjölund et al., 2015), mitigating image artifacts

from eddy currents (Yang and McNab, 2019) and concomitant gradients (Szczepankiewicz et al., 2019), and further minimizing side lobes in the encoding spectra (Hennel et al., 2020), we anticipate that the waveforms may be adapted for human in vivo studies. The merging of oscillating gradients (Aggarwal, 2020) and tensor-valued encoding (Reymbaut, 2020) into a common acquisition protocol encourages further development of a joint analysis framework, for instance, by augmenting current nonparametric diffusion tensor distributions (Topgaard, 2019b) with a Lorentzian frequency dimension (Narvaez et al., 2021, 2022) or building on the concept of confinement tensors (Yolcu et al., 2016; Boito et al., 2022).

**Code availability.** MATLAB code for image reconstruction, the definition of regions of interest, and curve fitting is available from https://github.com/daniel-topgaard/md-dmri (Topgaard, 2021; Nilsson et al., 2018).

**Data availability.** Experimental data are available from https:// github.com/daniel-topgaard/md-dmri-data (Topgaard, 2021).

**Supplement.** The supplement related to this article is available online at: https://doi.org/10.5194/mr-4-1-2023-supplement.

**Author contributions.** DT conceived the project. HJ, LS, and DT developed theory and software. HJ acquired data. HJ and DT processed and analyzed data. HJ, LS, and DT wrote the manuscript.

**Competing interests.** The contact author has declared that none of the authors has any competing interests.

**Disclaimer.** Publisher's note: Copernicus Publications remains neutral with regard to jurisdictional claims in published maps and institutional affiliations.

**Financial support.** This research has been supported by the Swedish Foundation for Strategic Research (Stiftelsen för Strategisk Forskning; grant no. ITM17-0267), the Swedish Research Council (Vetenskapsrådet; grant nos. 2018-03697 and 2022-04422_VR), and the China Scholarship Council.

**Review statement.** This paper was edited by Kong Ooi Tan and reviewed by Tom Barbara and one anonymous referee.

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

### Remarks from the language copy-editor

**CE1** Thank you for your feedback regarding the hyphenation throughout the paper. Please note, however, that the hyphenation has been adjusted to our house standards and cannot be altered. Our house standards have been developed in order to ensure homogeneity within and across our journals, and they conform to reputable reference standards such as The Chicago Manual of Style. If you feel that the hyphenation impacts the intended meaning in your specific paper, please provide us with reputable sources supporting this; we can then review instances on a case-by-case basis. Thank you.

### Remarks from the typesetter