# Peer review of "Multidimensional encoding of restricted and anisotropic diffusion by double rotation of the q-vector"

_Magnetic Resonance, 2022_

## Referee Comment (RC1)

Comments on MR-2022-16 "Multidimensional encoding of restricted....."

A review by Thomas M Barbara, AIRC, OHSU, Portland, OR.

This effort is the latest in a long standing and ongoing program from the Lund group on diffusion as studied by magnetic resonance. In agreeing to referee this paper I wanted to finally study this topic more than a casual glance given in the past. Even though I have read both of the monographs by Callaghan I do not consider myself to be an expert in this field.

Extending past work to gradient modulations that mimic double rotation used in solid state NMR appears to be something with promising potential for advancing the field. However, I did not find much motivation for this in the submission. DOR in solid state NMR of course has strong motivation because of the need to average out both of Legendre's second and fourth order polynomials. If something like that exists in the diffusion field, that aspect did not receive proper attention in this submission.

I also find myself in something of a conundrum regarding the theoretical exposition of this paper. This has to do with my understanding of the "b matrix" as a "rank one" matrix and therefore such a matrix will have very simple eigenvalues, namely (0,0,1) apart from a scaling factor (See attached notes). If such an understanding is correct, then the $b\_sub\_delta$ (equation 8) will have a very simple expression and the $b\_sub\_eta$ (equation 10) will be zero for any modulation and for any frequency spectrum of such modulation. Therefore Equation 31 of this submission seems to not be connected at all with Equation 4 and readers such as myself will be left in the dark as to what exactly is going on. From this perspective the submission is significantly lacking in clarity. Such clarity should be exposed on a general level, without requiring a reader to get into the details of this or that construction of a specific modulation scheme.

It could turn out that the conundrum with rank one matrices is a simple misinterpretation of notation, and I would be happy if that was the case, but I feel sufficiently confident to bring it up in print. The resolution will likely require a significant revision if that is the case.

- The matrix $b$ has elements

$$b_{ij} = g_i(\omega)\, g_j(-\omega) = g_i\, g_j^*$$

Thus $b$ is a rank one matrix and

$$b^2 = |\underset{\sim}{g}(\omega)|^2\, b = \left\{ \sum_k g_k(\omega)\, g_k^*(\omega) \right\} b$$

- Trace $b = |g|^2 \qquad \text{Tr}\, b^2 = |g|^4 = (\text{Tr}\, b)^2$

The characteristic equation for the $b$ matrix is therefore

$$\lambda^3 - (\text{Tr}\, b)\,\lambda^2 + \tfrac{1}{2}\left( \overset{0}{\overbrace{(\text{Tr}\, b)^2 - \text{Tr}\, b^2}} \right)\lambda - \overset{0}{\overbrace{\det b}} = 0$$

OR

$$\lambda^2 \left( \lambda - |g|^2 \right) = 0$$

thus $\qquad b_\Delta = \dfrac{bzz}{b} \qquad\qquad b_\eta = 0$

---

## Referee Comment (RC4)

**General Comments**

This is a very interesting manuscript and goes towards furthering the amount of information that will ultimately be available from clinical MRIs in the future.

Although the theory looks correct to me, I think that a more detailed discussion/elaboration of the mathematical derivations would make this work much more usable to the MRI audience and likely to increase its uptake and the likelihood of it filtering through to clinical usage. The MR literature is replete with examples of excellent theoretical developments being lost in the literature because the explanation of the theoretical developments in the papers were too brief – some of the works by Stepišnik come to mind. Following on from the comments of Tom Barbara, a more complete description, say as Supplementary Information, would really assist and avoid a "paper chase". And now, given the possibility of having large electronic Supplementary Information documents there is no reason not to.

**Specific Comments**

Lines 9-10  The sentence "Higher specificity to restriction … to the property not being of direct interest" is not easy to understand, at least on the first pass. The authors might like to rewrite this sentence.

Around lines 19, many of the intended readership would benefit from a clear explanation of what is meant by the frequency $\omega$ and how it relates to the pulse sequence parameters.

Eqs. (14) and (15), it might be advisable to use a letter different than $A$ since in Eq. (17) A is used in $D_A$ to signify axial.

Some of the Figures or parts of Figures are too small. (e.g., the lower two graphs in Figure 1, in Fig. 3c perhaps just plot a subset of the data).

Line 160. What was $T_1$ reduced to?

Line 164. What was actually done in allowing for the cell sedimentation? Was the supernatant removed?

Line 173. Did the 5 s recycle delay include the acquisition time?

The Proof of Principle experiments were conducted on high field very high gradient MRI equipment. It would be useful to see more discussion/outlook about the available parameter space (e.g., $\omega_{cent}$) and implications for running on clinical MRI equipment (e.g., lower gradient strengths) – assuming that SNR wasn't a limiting factor.

Line 206. Why didn't the extracellular compartments exhibit restricted ($\omega$-dependent) diffusion. Was it because the restrictions were not characterised by a reasonably homogeneous characteristic distance?

**Technical Corrections**

Line 77. axial and radial eigenvalues.
Line 88. Following previous works.

Line 99. Eq. (2)   that is, include a space before the "("/

---

## Author Response (AR1)

Dear Editor,

We thank the reviewers for their valuable comments and suggestions for changes. In general, both reviewers are happy with content and conclusions of the manuscript but request additional background material intended for a more general NMR and MRI audience not familiar with the details of the theory for frequency-domain analysis of diffusion encoding. In the revised version of the manuscript, we have taken into account all reviewer comments and, in particular, significantly expanded the background section and added a pedagogical figure. For consistency with the new material, we have throughout the text made some minor modifications that are not a direct response to any reviewer comment. With the exception of the completely new section, all changes are highlighted in the track-changes document.

Below follows detailed responses (preceded with "Authors' response:") to all reviewer comments which are included in full in italics.

Sincerely,

Daniel Topgaard

**RC1**: *'Comment on mr-2022-16'*, *Tom Barbara, 08 Nov 2022*
*Comments on MR-2022-16 "Multidimensional encoding of restricted....."*
*A review by Thomas M Barbara, AIRC, OHSU, Portland, OR.*
*This effort is the latest in a long standing and ongoing program from the Lund group on diffusion as studied by magnetic resonance. In agreeing to referee this paper I wanted to finally study this topic more than a casual glance given in the past. Even though I have read both of the monographs by Callaghan I do not consider myself to be an expert in this field. Extending past work to gradient modulations that mimic double rotation used in solid state NMR appears to be something with promising potential for advancing the field. However, I did not find much motivation for this in the submission. DOR in solid state NMR of course has strong motivation because of the need to average out both of Legendre's second and fourth order polynomials. If something like that exists in the diffusion field, that aspect did not receive proper attention in this submission.*
Authors' response: Although there are many formal similarities between solid-state and diffusion NMR, this comment highlights that the analogies shouldn't be taken too far. We have not yet found a diffusion property corresponding to second order quadrupolar broadening. Instead, the effect of DOR in diffusion is to remove the anisotropy in spectral content of the encoding tensor. Some further discussion on the similarities and differences between solid-state and diffusion was added on page 10, lines 204-210.

*I also find myself in something of a conundrum regarding the theoretical exposition of this paper. This has to do with my understanding of the "b matrix" as a "rank one" matrix and therefore such a matrix will have very simple eigenvalues, namely (0,0,1) apart from a scaling factor (See attached notes). If such an understanding is correct, then the b_sub_delta (equation 8) will have a very simple expression and the b_sub_eta (equation 10) will be zero for any modulation and for any frequency spectrum of such modulation. Therefore Equation*

*31 of this submission seems to not be connected at all with Equation 4 and readers such as myself will be left in the dark as to what exactly is going on. From this perspective the submission is significantly lacking in clarity. Such clarity should be exposed on a general level, without requiring a reader to get into the details of this or that construction of a specific modulation scheme.*

Authors' response: Although the conundrum was resolved in later comments, this comments highlights that the previous text was too focused on readers from the small sub-group of diffusion NMR developers familiar with the tensorial aspects of diffusion encoding. As a remedy, we have expanded the theory section with pedagogical material and an associated figure illustrating the origin of the non-zero values of the encoding anisotropy and asymmetry parameters b_Delta and b_eta.

**RC2**: *'Reply on RC1', Tom Barbara, 09 Nov 2022*
*Daniel cleared up my confusion by pointing out that all the gradients (imaging, diffusion, crusher, and slice selection) are included in the integral of Equation 4. That of course is not rank one. However, I believe that my confusion is shared by many, even though they are involved in diffuion studies in MRI and NMR, so having some language to make it clear from the outset is very much worth the effort. Indeed many of the references to the past literature merely repeat the stylized notation and so a reader new to the field cannot get the essential point very easily unless they know exactly where to look. I feel it is also important to include a diagram of the pulse sequence with all the gradients that are believed to contribute to this "sum over all gradients".*
*It could turn out that the conundrum with rank one matrices is a simple misinterpretation of notation, and I would be happy if that was the case, but I feel sufficiently confident to bring it up in print. The resolution will likely require a significant revision if that is the case.*

Authors' response: See previous reply about the significantly expanded and more pedagogical theory section and figure.

**RC3**: *'Reply on RC2', Tom Barbara, 15 Nov 2022*
*I forgot to mention that a very nice paper (also suggested by Daniel) is Mattiello,Basser and LeBihan, journal of magnetic resonance A108, 131 (1994). In that paper one can see that because of the discontinuous nature of the gradient time evolution, the integral breaks up into a finite sum of terms and the construction of the b matrix will contain rank one contributions from each term and each of the pairwise cross terms.*

Authors' response: The Mattiello et al paper indeed nicely illustrates the origin of the cross terms in the b-matrix that was introduced already in the preceding Basser, Mattiello, and Le Bihan paper in *J. Magn. Reson. B 103*, 247 (1994)

**RC4**: *'Comment on mr-2022-16', Anonymous Referee #2, 01 Jan 2023*
*General Comments*
*This is a very interesting manuscript and goes towards furthering the amount of information that will ultimately be available from clinical MRIs in the future. Although the theory looks correct to me, I think that a more detailed discussion/elaboration of the mathematical derivations would make this work much more usable to the MRI audience and likely to increase its uptake and the likelihood of it filtering through to clinical usage. The MR literature is replete with examples of excellent theoretical developments being lost in the literature because the explanation of the theoretical developments in the papers were too brief – some of the works by Stepišnik come to mind. Following on from the comments of Tom*

*Barbara, a more complete description, say as Supplementary Information, would really assist and avoid a "paper chase". And now, given the possibility of having large electronic Supplementary Information documents there is no reason not to.*

Authors' response: A more complete description of the theoretical background and an associated pedagogical figure was added in Section 2. Although the theory as such can be found in textbooks, the example with a 3D gradient modulation and random walk simulation has some novelty, and we choose to include the new material in the main text rather than in Supplementary Information.

*Specific Comments*

*Lines 9-10 The sentence "Higher specificity to restriction ... to the property not being of direct interest" is not easy to understand, at least on the first pass. The authors might like to rewrite this sentence.*

Authors' response: This sentence was rephrased for clarity (page 1, line 9-10).

*Around lines 19, many of the intended readership would benefit from a clear explanation of what is meant by the frequency $\omega$ and how it relates to the pulse sequence parameters.*

Authors' response: The meaning of the frequency domain is now more clearly explained in the new section 2.4.

*Eqs. (14) and (15), it might be advisable to use a letter different than A since in Eq. (17) A is used in $D_A$ to signify axial.*

Authors' response: The axial and radial diffusivities were renamed to parallel and perpendicular throughout, thus resolving the clash with the letter $A$ in Eq 44 (former Eq 17).

*Some of the Figures or parts of Figures are too small. (e.g., the lower two graphs in Figure 1, in Fig. 3c perhaps just plot a subset of the data).*

Authors' response: The details of an example gradient waveform are now more clearly visible in the new Figure 1a. The flowchart in fig 2 (former fig 1) is intended to summarize the steps in the calculation rather than showing the details of the waveforms and we prefer to keep the small size of the panels. The widths, grayscale, and order of the lines and symbols in fig 4c (former fig 3c) were modified to facilitate for the reader to visually observe the trend with frequency but not necessarily distinguish each individual datapoint.

*Line 160. What was $T_1$ reduced to?*

Authors' response: Information about T1 was added on page 11 line 352.

*Line 164. What was actually done in allowing for the cell sedimentation? Was the supernatant removed?*

Authors' response: More details on the yest sample preparation was added on page 11 lines 356-358 and page 13 lines 434-435. Most importantly, the image slice was placed in the middle of the cell sediment approx. 10 mm below the supernatant.

*Line 173. Did the 5 s recycle delay include the acquisition time?*

Authors' response: Following NMR jargon, the recycle delay just includes the time between the end of signal acquisition and the next excitation pulse.

*The Proof of Principle experiments were conducted on high field very high gradient MRI equipment. It would be useful to see more discussion/outlook about the available parameter space (e.g., ωcent) and implications for running on clinical MRI equipment (e.g., lower gradient strengths) – assuming that SNR wasn't a limiting factor.*

Authors' response: Conjecturally, for given hardware, b-value, and echo time, it should be possible to reach the same frequencies with spherical b-tensors as in more conventional linear oscillating gradients. The current version based on simple mathematical expressions is however only applicable to hardware with unusually high gradient capabilities and we prefer to postpone the discussion on the performance on more conventional equipment to future papers dealing with sequence optimization along the lines of our previous works (Topgaard, 2013; Sjölund et al., 2015).

*Line 206. Why didn't the extracellular compartments exhibit restricted (ω-dependent) diffusion. Was it because the restrictions were not characterised by a reasonably homogeneous characteristic distance?*

Authors' response: Indeed, any frequency dependence of the extracellular component may be masked by the structural disorder of the extracellular space which effectively includes the periplasm between the plasma membrane and the cell wall. The structural complexity and heterogeneity in chemical composition of the space accessible to the water outside the plasma membranes make it very challenging to predict the magnitude of this frequency dependence. In this manuscript, we simply note that the data is well fitted with a single isotropic Gaussian for the extracellular component while the pronounced effects of restriction is attributed to the intracellular one.

*Technical Corrections*
*Line 77. axial and radial eigenvalues.*
*Line 88. Following previous works.*
*Line 99. Eq. (2) that is, include a space before the "("/*
Authors' response: All corrections were implemented.

**RC5**: *'Reply on RC4', Tom Barbara, 03 Jan 2023*
*I was relieved to see that the second reviewer agreed with mine that the theory could use some clairification and I agree with the terminology used to point to the "paper chase" syndrome. I have experienced it many times. Supplementary information is a great idea and I would be happy to see it, but I want to emphasize that even a brief mention of the reality behand the stylized notation along with a citation to an accurate reference that gives further explaination, will go a long way in making readers happy and able to champion an effort as "a great paper". I do have coworkers who do work in the area of diffusion and I do talk to them (and my name often gets on some publication as a result) and very often I hear "well I could not understand that paper".*

Authors' response: These comments have been very useful for pointing out the weaknesses in the previous version of the manuscript having inordinate demands on the readers' familiarity with the theory of frequency-domain analysis of anisotropic diffusion and general gradient waveforms. The significantly expanded and more pedagogical theory section and figure will hopefully aid in reaching a broader NMR and MRI audience beyond the narrow circle of diffusion developers.